# Effector Sntf2 Interacted with Chloroplast-Related Protein Mdycf39 Promoting the Colonization of *Colletotrichum gloeosporioides* in Apple Leaf

**DOI:** 10.3390/ijms23126379

**Published:** 2022-06-07

**Authors:** Meiyu Wang, Zhirui Ji, Haifeng Yan, Jie Xu, Xuanzhu Zhao, Zongshan Zhou

**Affiliations:** Research Institute of Pomology, Chinese Academy of Agricultural Sciences, Xingcheng 125100, China; wang903108175@163.com (M.W.); xinyu_jzr@163.com (Z.J.); yanhaifeng50@163.com (H.Y.); szpyzt@163.com (J.X.); 72109019@mail.edu.com (X.Z.)

**Keywords:** *Colletotrichum*, effector, pathogenicity, immune response, apple

## Abstract

Glomerella leaf spot of apple, caused by *Colletotrichum*
*gloeosporioides*, is a devastating disease that leads to severe defoliation and fruit spots. The *Colletotrichum* species secretes a series of effectors to manipulate the host’s immune response, facilitating its colonization in plants. However, the mechanism by which the effector of *C. gloeosporioides* inhibits the defenses of the host remains unclear. In this study, we reported a novel effector Sntf2 of *C. gloeosporioides*. The transient expression of *S**NTF2* inhibits BAX-induced cell death in tobacco plants. Sntf2 suppresses plant defense responses by reducing callose deposition and H_2_O_2_ accumulation. *SNTF2* is upregulated during infection, and its deletion reduces virulence to the plant. Sntf2 is localized to the chloroplasts and interacts with Mdycf39 (a chloroplast PSII assembly factor) in apple leaves. The *Mdycf39* overexpression line increases susceptibility to *C. gloeosporioides*, whereas the *Mdycf39* transgenic silent line does not grow normally with pale white leaves, indicating that Sntf2 disturbs plant defense responses and growth by targeting Mdycf39.

## 1. Introduction

In agricultural and natural ecosystems, plants are exposed to numerous pathogens. Nevertheless, plants have complex defense systems to combat pathogen invasion [1]. When pathogen-associated molecular patterns (PAMPs) are recognized by pattern recognition receptors (PRRs) on the cell surface, the plant’s basal defense response is activated, termed PAMP-triggered immunity (PTI) [2]. Damage-associated molecular patterns (DAMPs)-triggered immunity (DTI) plays an important role in a plant’s basal defense response [3,4]. Plants have evolved intracellular nucleotide-binding leucine-rich-repeat receptors (NLRs) to recognize pathogen effectors directly or indirectly, leading to activation of the second line of defense, known as effector-triggered immunity (ETI) [2]. However, pathogens secrete a series of virulence effectors that interfere with the plant’s immune system, promoting pathogen invasion [5,6]. Hence, exploring the virulence mechanisms of effectors is important for revealing the infection strategies of plant pathogens.

The *Colletotrichum* species, belonging to Glomerellaceae of Ascomycota, develop penetration pegs from appressoria to invade the host [7]. *Colletotrichum* deploys distinct effectors at different infection phases to manipulate the host plant’s immune response [8]. Although various effectors have been identified [9,10], few have investigated their specific function mechanisms [11,12]. The interaction mechanism between effectors and host target proteins of *Colletotrichum* mainly focused on *Colletotrichum higginsianum*, *C**olletotrichum fruticola*, and *Colletotrichum*
*orbiculare* [13,14,15]. Each *Colletotrichum* species has evolved a set of effectors with unique strategies to adapt to its host plants [16]. In *C. gloeosporioides*, many studies focused on identifying the pathogenicity-related genes [17,18], but few focused on characterizing the mechanisms of effectors in the interactions between *C. gloeosporioides* and host plants.

*C. gloeosporioides* is a plant pathogenic fungus that infects various plants, including apple, mango, and poplar [19]. Glomerella leaf spot of apple (GLSA), caused by *C. gloeosporioides*, is a devastating disease that severely affects apple production [20,21]. Under favorable conditions, the latent period of this disease is as short as two days, and it spreads rapidly [19,22], causing necrotic fruit spots and severe defoliation in ‘Royal Gala’ and ‘Golden Delicious’ apples. Todate, the control of GLSA remains a challenge. Understanding the molecular interactions between *C. gloeosporioides* and apples is instrumental for sustaining effective disease control and developing disease-resistant varieties.

Chloroplasts not only play a role as photosynthetic organelles but also play a central role in plant defense by integrating environmental stimuli and the determinants of downstream defense responses [23,24,25]. Upon perception of pathogenic threat, chloroplast as the source of calcium, salicylic acid (SA), and reactive oxygen species (ROS) bursts communicates with the nucleus through retrograde signaling [24], mediates activation of plant immune signaling, and leads to the expression of defense-related genes [26,27]. To interfere with the function of chloroplasts in the interaction between the pathogens and the hosts, pathogens, including bacteria, viruses, fungi, and Oomycetes, have deployed effectors to target chloroplasts [6,23,27,28,29]. However, the underlying mechanisms with which effectors target chloroplasts to manipulate the host’s defenses have remained elusive in *Colletotrichum*.

In this study, we found that a novel effector Sntf2 of *C. gloeosporioides* played an essential role in suppressing the plant’s defense responses. Sntf2 is an extracellular-secreted protein including a signal peptide in its N-terminal, which plays a vital role in plant infection. Consistent with this, the deletion of *SNTF2* triggered the plant’s defense responses, in which H_2_O_2_ accumulation and callose deposition increased in the apple leaves inoculated with the Δ*sntf2*-1 mutant. We showed that Sntf2 could inhibit Bcl-2-associated X protein (BAX)-induced cell death, suppressing hypersensitive responses in plants. Our investigation demonstrated that Sntf2 could migrate into a host plant’s cells and interact with the Mdycf39 (a photosystem II assembly factor of *Malus domestica*). We also found that Mdycf39 played a vital role in plant resistance and development. Overall, our results show that Sntf2 perturbs the function of chloroplast, avoiding to trigger cell death and supporting pathogen colonization on live plants.

## 2. Results

### 2.1. Sntf2 Inhibits BAX-Induced Cell Death in Tobacco

BAX, a pro-apoptotic protein, stimulates cell death, which closely resembles the hypersensitive responsive (HR) in plants [30]. *SNTF2* (CGGC5_13909) encodes a putative protein in *C. gloeosporioides* with 187 amino acid residues and contains a predicted signal peptide (SP; 1–20 aa) (Figure 1a). To investigate if Sntf2 inhibited a plant’s innate immunity, we performed transient expression of the gene in *Nicotiana benthamiana*. The *Agrobacterium* strain GV3101 carrying the potato virus X (PVX) vector pGR106-fused *SNTF2* gene was infiltrated into the leaves of *N*. *benthamiana*, and the strain carrying pGR106-BAX was injected into the same sites on the leaves after 24 h post-infiltration. We found that Sntf2 suppressed the cell death triggered by BAX in the infiltrated leaves (Figure 1b). These data indicated that Sntf2 suppressed the HR response of the plant.

### 2.2. SNTF2 Is Upregulated during Biotrophic Infection Phase

To analyze the expression pattern of *SNTF2* at different infection phases, the transcripts of *SNTF2* were examined in infected leaves at 12, 24, 48, and 72 h post-inoculation (hpi) using qRT-PCR. *SNTF2* was upregulated at 24 hpi and 48 hpi (Figure 2). Appressoria formed at 12 hpi, and appressorium-mediated penetration occurred at 24 hpi (Appendix A). Fungal primary infection hyphae formed at 48 hpi (Appendix A). This meant that *SNTF2* was highly expressed from appressorium-mediated penetration to infection-hyphae formation. In addition, fluorescence signal detection showed that Sntf2-eGFP was actually expressed during the infection in apple leaves (Appendix A). These results suggest that *SNTF2* may play an important role in plant infection.

### 2.3. Sntf2 Is Required for the Pathogenicity of C. gloeosporioides

To determine the function of *SNTF2* in the pathogenicity, *SNTF2* deletion mutants were constructed using *A. tumefaciens*-mediated homologous recombination (Appendix A). The Δ*sntf2* mutants were confirmed by PCR and Southern blot analysis (Appendix A). The complementation strain Δ*sntf2*-1/*SNTF2* was generated and validated by PCR analysis (Appendix A). The Δ*sntf2* mutant had a normal growth rate compared with the wild type, as well as conidial production, appressorial formation, and invasive pegs formation (Appendix A). The apple leaves inoculated with the Δ*sntf2* mutants showed tiny spots compared with the WT (Figure 3), indicating that *Sntf2* was required for the pathogenicity of *C. gloeosporioides*.

### 2.4. Sntf2 Suppresses Apple Defense Responses

To evaluate if Sntf2 suppressed apple defense responses during the infection, we tested the generation of H_2_O_2_ and callose in apple leaves. The 3,3’-diaminobenzidine (DAB) staining result showed that H_2_O_2_ accumulation increased at the invasive sites when inoculated with the Δ*sntf2*-1 mutant compared with the WT (Appendix A). The H_2_O_2_ accumulation in the apple leaves inoculated with the complementation strain Δ*sntf2*-1/*SNTF2* was similar to the WT (Figure 4a). The rate of callose formation in apple leaves inoculated with the Δ*sntf2*-1 mutant was 24.4% higher than the leaves inoculated with the WT (Figure 4b). Aniline blue staining also showed that callose deposition in leaves inoculated with the Δ*sntf2*-1 mutant increased at 36 hpi compared with the WT (Appendix A). These results indicated that the deletion of *SNTF2* triggered apple defense responses.

### 2.5. Sntf2 Is a Secretion Protein

SignalP 4.0 analysis revealed that a SP was present in Sntf2 (SP; 1–20 aa; Figure 1a). To verify the function of this SP, an expression vector harboring the yeast invertase sequence fused to the SP was constructed and transformed into yeast strain YTK12. The transformant containing pSUC2-SP_SNTF2_ could grow on YPRAA plates (Figure 5a), which was consistent with the positive Avr1b [31], indicating that the SP of Sntf2 was able to secrete invertase. The invertase secreted by the transformant was confirmed by using 2, 3, 5-triphenyltetrazolium chloride (TTC) assays (Appendix A). To observe the extracellular secretion of Sntf2, we constructed a Sntf2-eGFP expression strain. We found that Sntf2-eGFP was secreted into plant cells (Figure 5b and Appendix A). These results revealed that the SP of Sntf2 was functional in mediating the extracellular secretion of Sntf2. Expectedly, the complementation mutant Δ*sntf2*-1*/SNTF2*^Δsp^ did not restore fully the pathogenicity of the Δ*sntf2*-1 strain (Appendix A), indicating that the SP of Sntf2 played a vital role in plant infection.

### 2.6. Sntf2 Is Localized to Plant Chloroplasts

To observe the subcellular localization of Sntf2, we transiently expressed the Sntf2^Δsp^-eGFP fusion protein using *A. tumefaciens* LBA4404 infiltration in *N*. *benthamiana*. We found that the fluorescence signal of Sntf2^Δsp^-eGFP fusion proteins overlapped with the chloroplast autofluorescence signal (Figure 6). The localization analysis showed that the Sntf2^Δsp^-eGFP was mainly presented in chloroplasts.

### 2.7. Sntf2 Interacts with the Photosystem II Assembly FactorMdycf39

To investigate Sntf2-interacting proteins, we performed a yeast two-hybrid (Y2H) screening assay. Twelve distinct putative interactors were confirmed via re-transformation into yeast (Appendix A and Appendix A). One of twelve positive genes encoded a chloroplast photosystem II assembly factor (MD05G1131800) (Appendix A). We designated the gene as *Mdycf39* since it was annotated as an ycf39-like protein in *Malus domestica* (NCBI databases). Mdycf39 is highly conserved with other ycf39-like proteins identified from plant species including *Arabidopsis thaliana* [32](80.76% identity to HCF244; Appendix A). The Mdycf39 contained a predicted chloroplast-targeting sequence (cTP; 27–64 aa) and a NAD(P)-binding domain (Figure 7a).

The yeast two-hybrid assay revealed that Mdycf39 interacted with Sntf2^ΔSP^ (lacking SP of Sntf2) (Figure 7b). We then transiently co-expressed the fuse proteins Sntf2^Δsp^-eGFP and Mdycf39-TagRFP in *N. benthamiana*. The Mdycf39-eGFP signal was detected in the chloroplasts, and the Mdycf39-TagRFP signal was co-localized with the signal of Sntf2^Δsp^-eGFP (Figure 7c). The bimolecular fluorescence complementation (BiFC) assay revealed that the fluorescence signal of the interaction between Sntf2-YFP^1−173^ and Mdycf39-YFP^173−238^ was present in chloroplasts (Figure 7d). The pull-down analysis also revealed that Sntf2^ΔSP^ interacted with Mdycf39 (Figure 7e). These results showed that Sntf2 interacted with the PS II assembly factor Mdycf39 in chloroplasts.

### 2.8. Mdycf39 Overexpression Increases Susceptibility to C. gloeosporioides in Apple

To study the function of *Mdycf39* in plant infection by targeting Sntf2, we generated two *Mdycf39* overexpression transgenic lines (OE-ycf39-1 and OE-ycf39-2) and a *Mdycf39* transgenic silent line (Ri-ycf39) (Appendix A). The OE-ycf39 lines showed no difference in plant growth compared with GL-3 (cv. Gala, a line) (Appendix A), but they showed increased susceptibility to *C. gloeosporioides* compared with GL-3 (Appendix A). However, the disease severity of the OE-ycf39 lines was similar to that of GL-3 when inoculated with Δ*sntf2*-1 (Figure 8). The result indicated that Sntf2 may suppress apple defenses responses by targeting Mdycf39. Unfortunately, the Ri-ycf39 line did not grow normally with palewhite leaves (Appendix A), which was consistent with the phenotype of *hcf244* mutant in *Arabidopsis* [32], resulting in the abortion of the Ri-ycf39 line in the pathogenicity test. These results indicated that *Mdycf39* played a vital role in plant resistance and development.

## 3. Discussion

The effector plays a vital role in manipulating the host’s immune responses and contributing to plant colonization by the pathogen [8,13,15]. In this study, we identified a novel effector Sntf2, which was required for the pathogenicity of *C. gloeosporioides*. *SNTF2* was highly expressed during the biotrophic infection phase. Effectors were induced to express and secreted to plant cells at different infection phases of Colletotrichum [33,34]. During the early biotrophic infection phase, effectors secreted to the plant cells interfered with immunity of the host, promoting the infection of the pathogen [34,35,36]. Indeed, in this study, we found that Sntf2 inhibited BAX-induced cell death, that the deletion of *SNTF2* induced callose deposition, and that H_2_O_2_ accumulation increased in infected leaves. These results indicated that Sntf2 played a role in suppressing plant defense and promoting biotrophic infection.

In this study, we found that Sntf2 interacted with Mdycf39 (a PSII assembly factor of *Malus domestica*) in chloroplasts. Mdycf39 is highly homologous to HCF244, a conserved PSII assembly factor in plants [32]. Mdycf39 and HCF244 are homologous to Ycf39 of *Synechocystis* [37], which was demonstrated to participate in synthesis of the D1 (a subunit of PSII reaction centre) and assembly of the PSII in chloroplasts [32,37]. The synthesis of the D1 was necessary to maintain the structure and function of the PSII [38]. These suggested that Mdycf39 may also be involved in the assembly of the PSII in chloroplasts of apples. In this study, the apple Mdycf39-RNAi line did not grow normally on MS, consistent with the phenotype of *hcf244* mutant in Arabidopsis [32]. These results indicated that Mdycf39 may affect the stability of PSII and play a role in plant growth and development.

Different suborganelles in the host often become targets of pathogens [23,39]. Many effectors target different components of the PSII in the host, which interferes with electron transport and inhibits ROS production, interfering with plant defense [6,28,40]. The effector RXLR31154 of *Plasmopara viticola* targets PsbP (a factor of the oxygen-evolving complex of PSII) to inhibit H_2_O_2_ production in a grapevine and enhance its susceptibility [6]. In our study, we found that the effector Sntf2 of *C. gloeosporioides* migrated into the host’s chloroplasts and interacted with Mdycf39. The transgenic overexpression lines of *Mdycf39* in GL-3 showed an increased susceptibility to *C. gloeosporioides*. Our study indicated that Mdycf39 plays an important role in plant photosynthesis and defense responses. Mdycf39 may affect the stability of PSII and the production of H_2_O_2_ in chloroplasts.The defense response of plants is usually associated with ROS generation and callose depositions [41]. ROS has a direct anti-microbial role and is also a retrograde signaling molecule entering the nucleus to regulate the expression of defense-related genes, resulting in hypersensitive cell death [42,43,44]. In this study, the deletion of *SNTF2* increased the accumulation of H_2_O_2_ and the deposition of callose during infection. We speculated that Sntf2 perturbs the function of chloroplast by targeting Mdycf39, inhibiting the apple’s defense response.

In conclusion, we identified a novel effector Sntf2 of *C. gloeosporioides*. Sntf2 is required for pathogenicity and plays a vital role in plant infection. The deletion of *SNTF2* triggers plant defense responses. Sntf2 secreted into the host cells was located in the chloroplast. Sntf2 interacted with Mdycf39 (a photosystem II assembly factor) in chloroplasts. We demonstrated that Sntf2 perturbs the function of chloroplasts by targeting Mdycf39, avoiding hypersensitive cell death, and supporting the colonization of *C. gloeosporioides* in apple leaves.

## 4. Materials and Methods

### 4.1. Strains and Plant Materials

The *C. gloeosporioides* strain W16 was used as the wild type (WT) strain [19]. The fungal strains were cultured on PDA plates at 26 °C as described previously [19]. For transformant selection, G-418 sulphate or hygromycin B was added to PDA infinal concentrations of 500 μg/mL or 100 μg/mL, respectively [45]. The tissue-cultured ‘GL-3′ (cultivar ‘Royal Gala’) plants [46] and the transgenic plants were cultivated in MS medium [47] in a climate-controlled culture room at 25 ± 1 °C with a 16/8 h light/dark photoperiod as described by Dai et al. [46]. *N. benthamiana* seedlings were cultured in a greenhouse at 22–25 °C. The healthy leaves were obtained from 2-year-old seedlings of the ‘Golden Delicious’ variety of apples (*Malus domestica*) (Institute of Pomology of Chinese Academy of Agricultural Sciences, CAAS, Xingcheng, Liaoning Province, China).

### 4.2. Agrobacterium tumefaciens Infiltration Assays

The coding sequence of *SNTF2* (CGGC5_13909) was amplified and ligated into the potato virus X (PVX) vector pGR106 to generate pGR106-SNTF2. The primers used in this assay are listed in Appendix A. The recombinant vector was transformed into *A. tumefaciens* GV3101. For transient expression in *N. benthamiana*, the transformant strains of GV3101 were infiltrated into the leaves. The experiment was performed as described by Shang et al. [15].

### 4.3. Vector Construction and Fungal Transformation

The gene deletion construction and transformation of *C. gloeosporioides* were carried out using the protocols described previously [45]. The gene complementation construction and the fungal transformation were performed as described previously [48]. The primers used in this assay are listed in Appendix A. A detailed description is shown in Appendix A. The putative gene knockout mutants were identified by PCR and Southern blot analysis. The complementation strains were confirmed based on PCR analysis.

### 4.4. Phenotype Assays

The hyphal growth rate and conidial production were assessed according to the method described by Zhou et al. [45]. The formation rates of appressorium and invasive pegs were observed and calculated as described by Shang et al. [15]. For plant inoculation, fresh conidial suspensions (1 × 10^5^ conidia/mL) were sprayed onto the apple leaves according to the method described by Zhou et al. [45]. The inoculated apple seedlings (2-year-oldseedlings of the ‘Golden Delicious’ variety as shown in Section 4.1) were cultured at 28 °C and 75% humidity. For the pathogenicity test, the disease lesions were examined 3 days post-inoculation. The severity of the GLSA on each leaf was estimated using a diagrammatic scale [49].

### 4.5. RNA Extraction and qRT-PCR Analysis

The total RNA was extracted using an RNAprep Pure Plant Kit (TianGen Biotech, China Beijing). For the qRT-PCR analysis of *SNTF2*, the RNA samples of the infestation phase were obtained from ‘Golden Delicious’ leaves (2-year-old seedlings of the ‘Golden Delicious’ variety, inoculated as shown in Section 4.4) at 12, 24, 48, and 72 hpi. For the qRT-PCR analysis of *Mdycf39*, the apple RNA samples were extracted from the transgenic overexpression lines of *Mdycf39* and tissue-cultured ‘GL-3′ plants. The qRT-PCR was performedas described by Tan et al. [17]. The *M. domestica* ubiquitin extension (*MdUBQ*) was used as the endogenous reference gene [15].

### 4.6. Signal Peptide Activity Assay

The SP sequence of *SNTF2* was amplified and ligated into pSUC2 vector to generate pSUC2-SP_SNTF2_. Recombinant vectors were transformed into the yeast strain YTK12, which lacks a secreted invertase [50]. All transformants were cultured on YPDA plates at 30 °C and cultured on CMD-W and YPRAA plates to assess if the invertase secreted. The enzyme activity of the invertase was evaluated based on the reduction of TTC to the insoluble red compound 1, 3, 5-triphenylformazan. The experiment was performed as described by Xu et al. [28].

### 4.7. Histochemical Assays

The H_2_O_2_ accumulation in plants was assessed using DAB staining as described by Chen et al. [51]. The samples were from ‘GL-3′ and transgenic leaves inoculated with the WT, Δ*sntf2*-1, and Δ*sntf2*-1*/SNTF2* strains. DAB oxidation leads to brownish polymer formation and deposition at the site of ROS accumulation. The callose deposition was observed using aniline blue staining. The samples were decolourized by boiling in 96% ethanol for 5 min and then immersed in chloral hydrate overnight. The transparent leaf segments were stained with 0.05% aniline blue in 0.067 M K_2_HPO_4_ (pH 9.6) [28]. Then processed samples were preserved in 30% glycerol for microscopic analysis (Leica DM5000 B, Leica, Wetzlar, Germany). For every time point, either ten leaf discs were processed using DAB staining or aniline blue staining was obtained for ten leaf discs, and every experiment was performed three times. For every leaf disc, 100 appressoria were observed, and the percentage of ROS accumulation or callose deposition was calculated along with the means and standard deviations.

### 4.8. Yeast Two-Hybrid Assay

The coding sequences of *SNTF2* (without SP) were cloned into pGBKT7-BD as the bait, and the Matchmaker GAL4 system (OE Biotech, Shanghai, China) was used to screen a cDNA library constructed from RNA and isolated from different infection phases of the ‘Golden Delicious’ apple leaves. The screening was performed according to the manufacturer’s instructions (OE Biotech, Shanghai, China). To confirm the interaction, the prey vector pAD-Sntf2^Δsp^ and bait vector pBD-Mdycf39 (constructed using *Mdycf39* from ‘Golden Delicious’) were co-transformed into Y2Hgold yeast strains. The transformed yeast strains were grown on a medium (SD/-Leu/-Trp and SD/-Leu/-Trp/-His/-Ade) at 30 °C.

### 4.9. Transient Expression Analysis in N. benthamiana

For the subcellular localization assay, *SNTF2* (without SP) and *Mdycf39* were cloned into pGR35s-eGFP or pGR35s-TagRFP and transformed into *A. tumefaciens* LBA4404 to express Sntf2^Δsp^-eGFP, Mdycf39-eGFP, and Mdycf39-TagRFP fusion proteins, respectively. The transient expression in *N. benthamiana* was performed as described by Xu et al. [28]. The *N. benthamiana* leaves were observed using a confocal microscope (Leica TCS SP8, Leica, Wetzlar, Germany) after 2 days of infiltration. For the BiFC assay, *SNTF2* and *Mdycf39* were ligated into the vectors pGR35s-YFP^1−173^ and pGR35s-YFP^173−238^. The Mdycf39-YFP^173−238^ and Sntf2^Δsp^-YFP^1−173^ fusion proteins were co-expressed by *A. tumefaciens* infiltration in *N. benthamiana*. The experiment was performed as described by Xu et al. [28]. GFP fluorescence was excited using a 488 nm laser, and emission was collected between 505 and 535 nm. YFP was excited using a 514 nm laser, and emission was collected between 530 and 560 nm. TagRFP was excited using a 555 nm laser, and emission was collected between 578 and 610 nm. For autofluorescent chloroplast detection, the excitation was 630 nm, and the collection range of emitted light was set at 650–681 nm [52].

### 4.10. Protein Extraction and Immunoblotting

For the pull-down assay, *SNTF2* and *Mdycf39* were cloned into pQE30-eGFP and pQE30-Flag and transformed into *Escherichia coli* M15 cells. Protein extraction was performed according to Dominguez-Martin et al. [53]. The induced proteins were observed using SDS-PAGE and Coomassie Brilliant Blue staining. The purified proteins were co-incubated on ice for 3 h and were purified using Ni-NTA beads (CW0894S, CWBIO, Beijing, China). For immune detection, the purified proteins were transferred to a polyvinylidene fluoride membrane. The corresponding mouse anti-His or mouse anti-Flag (1:1000; cat. no. HT201, TransGenBiotech, Beijing, China) antibody was used as the primary antibody, respectively. The HRP-labelled goat anti-mouse IgG (1:2000; cat. no. A0216, Beyotime, Shanghai, China) was used as the secondary antibody. The membrane was treated with a BeyoECL Star kit (Beyotime, Shanghai, China) for 2 min. Images were acquired using a BIO-RAD ChemiDoc™ Imaging System.

### 4.11. Generation of Transgenic ‘GL-3’Plants with Mdycf39 Overexpression or RNA-Interference

*Mdycf39* was amplified and inserted into the pRNAi and pRPHA vectors to generate the RNAi transgenic silenced line Ri-ycf39 as well as the transgenic overexpression lines OE-ycf39. The *Agrobacterium*-mediated transformation of ‘GL-3’ was performed as described by Dai et al. [46]. *A. tumefaciens* LBA4404 was used for the stable transformations.

## Figures and Tables

**Figure 1 ijms-23-06379-f001:**
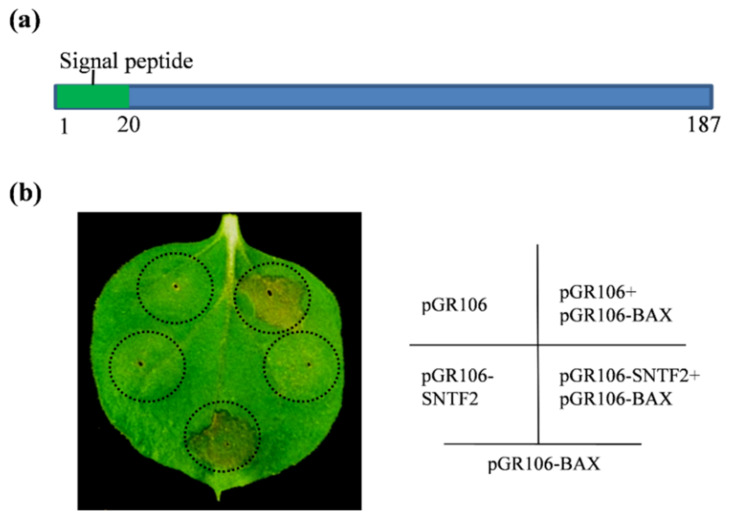
Sntf2 suppressed BAX-triggered cell death in *Nicotiana benthamiana*. (**a**) The structure analysis of Sntf2. Sntf2 was predicted to contain an N-terminal signal peptide (1–20 aa). (**b**) The tobacco leaves infiltrated with *Agrobacterium*
*tumefaciens* GV3101 harboring the pGR106 vector or the vector with *SNTF2* inserted. The *A. tumefaciens* carrying the *BAX* gene were injected into the leaves after 24 h post-infiltration. The empty pGR106 vector was used as the control. The black dotted lines indicate the region of infiltration. The diagram on the right shows the transformant strains of *A. tumefaciens* (containing recombinant pGR106 vectors) that was injected inside the black dotted lines. Images were acquired at 7 days post-infiltration. The experiments were repeated six times.

**Figure 2 ijms-23-06379-f002:**
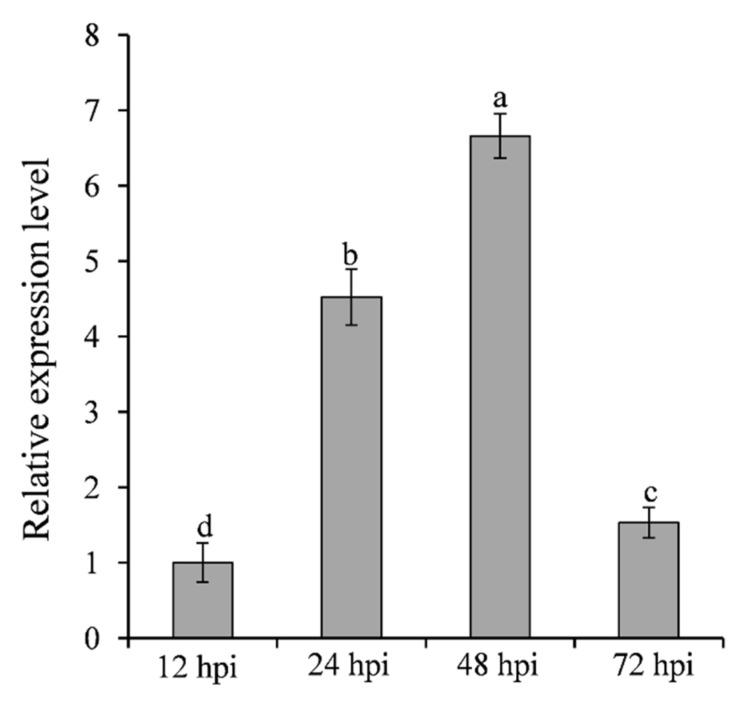
Analysis of *SNTF2* transcription at different infection phases by qRT-PCR. The cDNA was obtained from the apple leaves at 12, 24, 48, and 72 h post-inoculation (hpi) with *Colletotrichum gloeosporioides*. The *M. domestica* ubiquitin extension factor (*MdUBQ*) gene was used as the reference gene to normalize and analysis the transcription of *SNTF2*. Results were presented as the average fold values from three independent experiments compared with 12 hpi samples. Error bars represent standard deviations. Lowercase letters indicate significant differences (*p* < 0.01).

**Figure 3 ijms-23-06379-f003:**
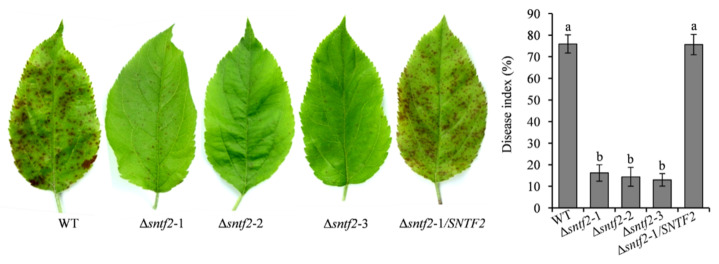
*SNTF2* deletion reduced the pathogenicity of *Colletotrichum gloeosporioides*. The pathogenicity was evaluated based on the disease index. Each experiment was repeated three times with ten leaves used for each replicate. Error bars represent standard deviations. Lowercase letters indicate significant differences (* p*< 0.01).

**Figure 4 ijms-23-06379-f004:**
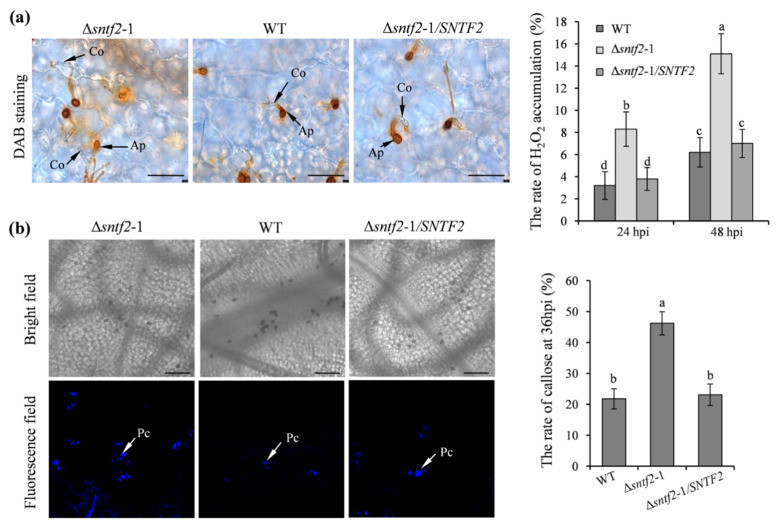
The H_2_O_2_ accumulation and callose deposition in apple leaves during the infection. (**a**) DAB staining was performed to detect H_2_O_2_ accumulation on apple leaves. DAB oxidation led to brownish polymer formation that was deposited at the site of H_2_O_2_ accumulation. Co: conidia; Ap: appressorium; bar: 50 μm. (**b**) Aniline blue staining was to observe callose deposition on apple leaves at 36 hpi. Apple leaves were inoculated with WT, Δ*sntf2*-1, Δ*sntf2*-2, Δ*sntf2*-3, or Δ*sntf2*-1/*SNTF2* strains. Pc: callose deposition; bar: 50 μm. The rate of H_2_O_2_ accumulation and callose formation was evaluated. Error bars represent standard deviations. Lowercase letters represent significant differences (*p* < 0.01).

**Figure 5 ijms-23-06379-f005:**
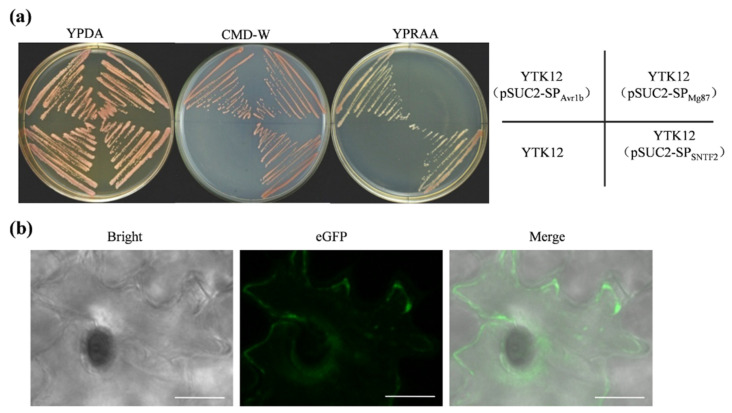
The secretory function validation of Sntf2. (**a**) The secretory function test of the predicted signal peptide (SP) of Sntf2 was based on the yeast’s secretory system. The YTK12 strains containing pSUC2-SP_Avr1b_ (SP_Avr1b_: the SP sequence of Avr1b from *Phytophthora sojae*) were used as a positive control, and those containing pSUC2-SP_Mg87_ (SP_Mg87_: the first 25 amino acids of Mg87 protein from *Magnaporthe*
*oryzae*) were used as negative controls. (**b**) Sntf2-eGFP fusion protein was translocated into plant cells during apple leaf infection. The leaves were inoculated with Δ*sntf2*-1/*gpdAp:SNTF2:eGFP* strain (*gpdAp*: the promoter of glyceraldehyde-3-phosphate dehydrogenase gene from *Aspergillus nidulans*). Bar: 10 μm.

**Figure 6 ijms-23-06379-f006:**
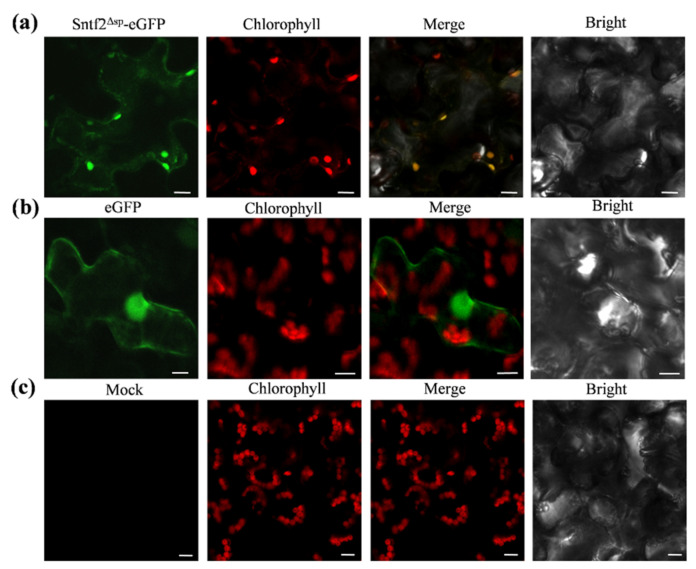
Sntf2^Δsp^-eGFP was localized to chloroplasts in *Nicotiana benthamiana*. (**a**) Sntf2^Δsp^-eGFP fluorescence signal overlapped with the chloroplasts autofluorescence signal. (**b**) The GFP protein was used as a control. (**c**) Mock. Bar: 10 μm.

**Figure 7 ijms-23-06379-f007:**
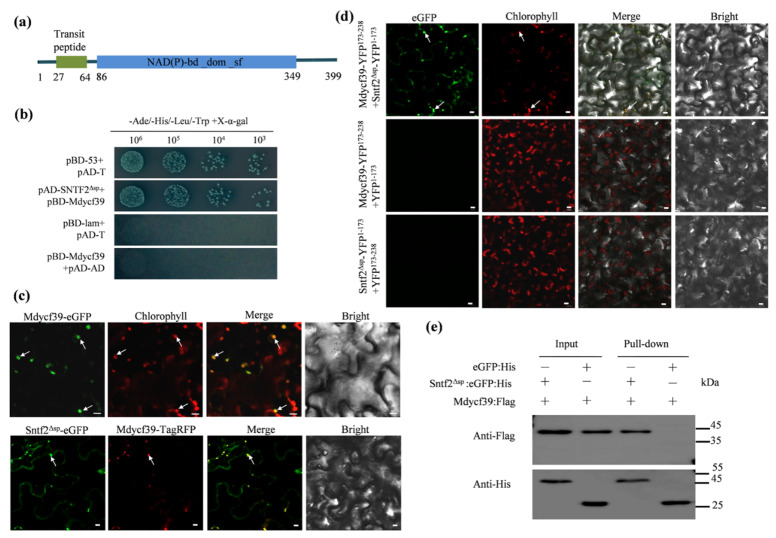
Sntf2 interacted with Mdycf39 in chloroplasts. (**a**) Mdycf39 was predicted to contain a chloroplast-targeting transit peptide and a NAD(P)-binding domain by the LOCALIZER program and InterPro 85.0. (**b**) Confirmation of the Sntf2–Mdycf39 interaction using yeast two-hybrid assays. (**c**,**d**) Detection of the Sntf2–Mdycf39 interaction using co-localization and bimolecular fluorescence complementation assays. The arrows refer to the chloroplasts. Bar: 10 μm. (**e**) Confirmation of Sntf2–Mdycf39 interaction using pull-down assays. Western blots for the analysis of Sntf2:eGFP:His and Mdycf39:Flag purified proteins from *Escherichia coli* M15 and proteins from the His purification column using anti-His and anti-Flag antibodies. The Sntf2^Δsp^:eGFP:His and Mdycf39:Flag bands were 46 and 45 kDa, respectively. The protein marker is labeled on the right.

**Figure 8 ijms-23-06379-f008:**
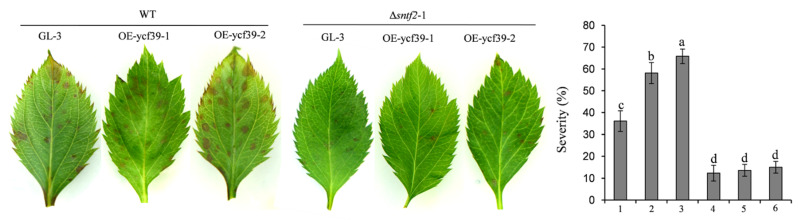
Analysis of the susceptibility of OE-ycf39 lines. The susceptibility was evaluated at three days after inoculation with WT and Δ*sntf2*-1 based on the disease index. 1–3: GL-3, OE-ycf39-1, and OE-ycf39-2 lines were inoculated with WT; 4–5: GL-3, OE-ycf39-1, and OE-ycf39-2 lines were inoculated with Δ*sntf2*-1. Three replicates were performed for each experiment, with six leaves for each replicate. Error bars represent standard deviations. Lowercase letters represent significant differences (*p* < 0.01).

## Data Availability

The data presented in this study are available in the article.

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
