# Peer review of "Effector Sntf2 Interacted with Chloroplast-Related Protein Mdycf39 Promoting the Colonization of Colletotrichum gloeosporioides in Apple Leaf"

_ijms, 2022, doi:10.3390/ijms23126379_

Round 1

Reviewer 1 Report

Comments and suggestions attached

Author Response

Line 29: please, add Damage-associated molecular patterns (DAMPs) and Damage Triggered Immunity(DTI) induced by cell wall degrading enzymes as basal defense layer. Several recent research papers arehighlighting this defensive strategy as important as PTI (i.e., see Zarattini et al, 2021 Comm Biol; Kubiceket al., 2014 Annual Review phytopath).

Answer: We have added the content and the reference in the revised manuscript, thanks for your suggestion.

Line 38: better introduce Colletotrichum, for example by explaining the type of fungi, ascomycete, lifestyle etc.

Answer: We have added a brief introduce for Colletotrichum in the revised manuscript.

Line 68: extracellular secreted protein

Line 72: every time you use an acronymous please report the full-length name during the firstappearance in the text, for example, Bcl-2-associated X protein (BAX)-associated cell death…

Line 74: when you refer to ycf39 protein, please report Md (Malus domestica) in Italic across all themanuscript, also in the title.

Answer: Thanks for your suggestion. We have revised it in the manuscript.

Line 83-84: please better explain the experimental method carried out, specify whether you performed a transient transformation, co-infiltration, or a stable transformation.

Answer: Thanks for your suggestion. We performed the transient expression of SNTF2 and BAX gene using Agrobacterium tumefaciens co-infiltration in tobacco leaves. We also have revised the experimental method in the revised manuscript.

Fig. 2: I would suggest the authors to split Fig. 2 in Fig. 2a (qRT-PCR related to hyphae and conidia) and Fig. 2b related to “in-planta” assay.

Answer: Thanks for your suggestion. We gave up the qRT-PCR related to hyphae and conidia, and only showed the qRT-PCR related to “infected leaves at 12, 24, 48, and 72 h post-inoculation (hpi)” assay.

Legend Fig. 2: please better explain how the relative expression data were normalized: how did you normalize the “in-planta” assay experiment? The fungal gene expression assessed during plant-pathogen interaction studies is generally normalized by using a plant housekeeping gene

Answer: The expression of SNTF2 was assessed during plant-pathogen interaction studies using MdUBQ (a ubiquitin extension factor gene of Malus domestica) gene for normalization.

Fig. legend 4: line 139, DAB staining was carried out to detect… moreover, the DAB staining is leading to brownish polymer formation, not yellow.

Fig. legend 5a is reported twice in the manuscript (same scheme as the Fig. 1a: please remove it from the supplementary material and cite the Fig. 1a in the text).

Answer: Thanks for your suggestion. We have revised it in the manuscript.

Materials and Methods

4.1: The age of plants used to perform plant-pathogen interaction studies is not reported, please specify

4.2: Please specify the code of the SNTF2 coding sequence

4.4: Line 303: Please report the citation number following Zhou et al. Please also specify the age of the plants used to perform phenotype assays

4.5: same as above, specify the age of plant material used

4.7: Line 335: DAB oxidation is brown.

Answer: Thanks for your suggestion. We have revised it in the manuscript. Healthy leaves for plant-pathogen interaction studies were obtained from two years seedlings of the ‘Golden Delicious’ variety of apples (Malus domestica) from the research orchard (Institute of Pomology of Chinese Academy of Agricultural Sciences, CAAS, Xingcheng, Liaoning Province, China). The code of SNTF2 coding sequence is CGGC5_13909, we have added note in the manuscript.

Reviewer 2 Report

The manuscript “Effector Sntf2 targets chloroplasts protein Mdycf39 and pro-2 motes Colletotrichum gloeosporioides colonization on apple leaf” reports the biological identification and characterization of fungal effector with clear evidence of its function on the pathogenicity. This approach is very interesting and enlighten how the fungal controls the plant defence mechanisms.

The manuscript is written clearly and understandably, and the conclusion are supported in the results presented.

My only concern is the use of only one reference gene in the RT-qPCR analysis.

Minor corrections are indicated along the manuscript.

Author Response

  1. My only concern is the use of only one reference gene in the qRT-PCR analysis.

Answer : MdUBQ, a ubiquitin extension gene of Malus domestica has confirmed that can be used as the reference gene to normalize and analysis the transcription of apple leaf genes (1).

(1) hang, S.; Wang, B.; Zhang, S.; Liu, G.; Liang, X.; Zhang, R.; Gleason, M. L.; Sun, G., A novel effector CfEC92 of Colletotrichum fructicola contributes to glomerella leaf spot virulence by suppressing plant defences at the early infection phase. Molecular Plant Pathology 2020, 21, (7), 936-950.

  1. Minor corrections are indicated along the manuscript.

Answer: We have checked the contents in the manuscript, and we have some revisions in the revised manuscript. The changes were marked with highlight in the revised manuscript.